# Evaluation of the 8th Edition TNM Classification for Anaplastic Thyroid Carcinoma

**DOI:** 10.3390/cancers12030552

**Published:** 2020-02-27

**Authors:** Naoyoshi Onoda, Iwao Sugitani, Ken-ichi Ito, Akifumi Suzuki, Takuya Higashiyama, Tatsuya Fukumori, Nobuyasu Suganuma, Katsuhiko Masudo, Hirotaka Nakayama, Atsuhiko Uno, Katsunari Yane, Seiichi Yoshimoto, Aya Ebina, Yukari Kawasaki, Shigeto Maeda, Manabu Iwadate, Shinichi Suzuki

**Affiliations:** 1Anaplastic Thyroid Carcinoma Research Consortium of Japan, Tokyo 113-8603, Japan; isugitani@nms.ac.jp (I.S.); kenito@shinshu-u.ac.jp (K.-i.I.); af-suzuki@ito-hospital.jp (A.S.); higashiyama@kuma-h.or.jp (T.H.); fukumori1024@kanaji.jp (T.F.); n-suga@vesta.dti.ne.jp (N.S.); masudo@yokohama-cu.ac.jp (K.M.); hirnak@yokohama-cu.ac.jp (H.N.); auno@gh.opho.jp (A.U.); kyane@med.kindai.ac.jp (K.Y.); seyoshim@ncc.go.jp (S.Y.); aya.ebina@jfcr.or.jp (A.E.); y-zenke@umin.ac.jp (Y.K.); maeda.shigeto.sr@mail.hosp.go.jp (S.M.); iwadate@fmu.ac.jp (M.I.); shsuzuki@fmu.ac.jp (S.S.); 2Department of Breast & Endocrine Surgery, Osaka City University Graduate School of Medicine, Osaka 545-8585, Japan; 3Department of Endocrine Surgery, Nippon Medical School, Tokyo 113-8603, Japan; 4Division of Breast and Endocrine Surgery, Department of Surgery, Shinshu University School of Medicine, Matsumoto 390-8621, Japan; 5Department of Surgery, Ito Hospital, Tokyo 150-8308, Japan; 6Department of Surgery, Kuma Hospital, Kobe 650-0011, Japan; 7Department of Surgery, Kanaji Thyroid Hospital, Tokyo 114-0015, Japan; 8Department of Breast and Endocrine Surgery, Kanagawa Cancer Center, Yokohama 241-8515, Japan; 9Department of Breast and Endocrine Surgery, Yokohama City University Medical Center, Yokohama 232-0024, Japan; 10Department of Surgery, Yokohama City University, Yokohama 236-0004, Japan; 11Department of Otolaryngology-Head and Neck Surgery, Osaka General Medical Center, Osaka 558-8558, Japan; 12Department of Otolaryngology-Head and Neck Surgery, Kindai University Nara Hospital, Ikoma 630-0293, Japan; 13Department of Head and Neck Surgery, National Cancer Center Hospital, Tokyo 104-0045, Japan; 14Department of Head and Neck Surgery, Cancer Institute Hospital, Tokyo 135-8550, Japan; 15Department of Surgery, Tsuchiya General Hospital, Hiroshima 730-0811, Japan; 16Department of Surgery, National Hospital Organization Nagasaki Medical Center, Omura 856-8562, Japan; 17Department of Thyroid and Endocrinology, Fukushima Medical University School of Medicine, Fukushima 960-1247, Japan

**Keywords:** anaplastic thyroid cancer, prognosis, disease stage, TNM classification, prognostic index

## Abstract

Background: The tumor–node–metastasis (TNM) classification system to categorized anaplastic thyroid cancer (ATC) was revised. Methods: The revised system was evaluated using a large database of ATC patients. Results: A total of 757 patients were analyzed. The proportion and median overall survival values (OS: months) for each T category were T1 (*n* = 8, 1.1%, 12.5), T2 (*n* = 43, 5.7%, 10.9), T3a (*n* = 117, 15.5%, 5.7), T3b (*n* = 438, 57.9%, 3.9), and T4 (*n* = 151, 19.9%, 5.0). The OS of the N0 and N1 patients were 5.9 and 4.3, respectively (log-rank *p* < 0.01). Sixty-three (58.3%) patients migrated from stage IV A to IV B by revision based on the existence of nodal involvement and 422 patients (55.7%) were stratified into stage IV B, without a worsening of their OS (6.1), leaving 45 patients (5.9%) in stage IV A with fair OS (15.8). The hazard ratios for the survival of the patients of stage IV B compared to stage IV A increased from 1.1 to 2.1 by the revision. No change was made for stage IV C (*n* = 290, 38.8%, 2.8). Conclusion: The revised TNM system clearly indicated the prognoses of ATC patients by extracting rare patients with fair prognoses as having stage IV A disease and categorized many heterogeneous patients in stage IV B.

## 1. Introduction

Anaplastic thyroid cancer (ATC) is a rare orphan disease that is refractory to therapeutic efforts. Its reported incidence is 1–4% of all thyroid cancer cases. More than 80% of ATC patients already have disease progression to the surrounding organs and/or distant metastasis at their initial presentation [1,2,3], and such extraordinarily rapid disease progression sometimes precludes the initiation of therapeutic attempts. In cases in which effective management cannot be provided, it is not unusual for an individual with ATC to die within several days of receiving the diagnosis. Compared to differentiated thyroid cancer (DTC), ATCs tend to develop in older individuals, and older patients may not be able to tolerate aggressive anticancer treatments due to deteriorations in their immune and/or organ functions and comorbidity. It can thus be very difficult to achieve a successful treatment for ATC [2].

Two studies demonstrated that the prognosis of ATC patients can be clearly estimated by the American Joint Committee on Cancer/Union for International Cancer Control tumor–node–metastasis (AJCC/UICC TNM) classification system (7th edition). Sugitani et al. used a Japanese database of 677 patients with ATC [1], and Haymart et al. analyzed 2742 patients from the United States National Cancer Database [4]. The results of both studies showed similar prognoses; the median overall survival (OS) of the ATC patients was 4 months for all patients, and 6, 4, and 2 months for stage IV A, IV B, and IV C patients.

The 7th edition of the TNM classification stratified ATC patients simply by the existence of extrathyroidal extension of the primary tumor and distant metastasis. All ATC patients were thus classified as having a T4 tumor and stage IV disease. ATCs confined to the thyroid gland were classified as T4a tumors, and stage IV A, and ATCs that extended beyond the thyroid capsule were stratified as T4b and stage IV B. Patients with distant metastasis were classified as stage IV C irrespective of their tumor and nodal status [5]. In the revised 8th edition, ATC tumors are classified as the same T category as DTCs [6]. All ATC patients are still classified into stage IV, as in the 7th edition. No further changes in the stage stratification according to the tumor are made by the 8th edition; instead, the presence of nodal metastasis is regarded as a factor that moves the patient into stage IV B **(**Table 1).

We conducted the present study to compare the prognoses of ATC patients by applying the 7th and 8th editions of the TNM classification, using a large cohort of ATC patients whose cases were accumulated in a nationwide database of the Anaplastic Thyroid Carcinoma Research Consortium of Japan (ATCCJ) [1] to investigate the influence of the 8th edition’s revision on the patients’ estimated prognoses.

## 2. Results

From the initial 1265 cases, 508 cases were excluded. As illustrated in Figure 1, 313 cases had ATC at the site of a metastatic or recurrent lesion only; 63 cases had an ATC lesion that was incidentally identified after the pathological investigation of the surgical specimen; 16 cases lacked prognosis information; 37 cases lacked stage category data; 22 cases lacked the data necessary to determine the T category by the 8th edition, and 57 cases lacked N-category data. Incidentally identified ATC was excluded because they often associated with concomitant papillary carcinoma and showed a more favorable prognosis compared with ordinally ATC [7]. Moreover, in those case, (1) accurate diameter of ATC might be difficult to measure, (2) nodal involvement and/or distant metastasis could often be from differentiated carcinoma, (3) preoperative or operative findings might be due to coexisting DTC and be inappropriate to describe the characteristics of ATC. Furthermore, some of these were found only within the metastatic lesion.

As summarized in Table 2, a final total of 757 patients was analyzed: 305 men and 452 women with a median age of 71.0 (28–96) years. The median size of the maximal tumor diameter was 6.0 (1–17) cm.

The OS for all 757 patients was 4.8 (95% CI: 4.3–5.4) months. In the 8th edition of the TNM classification, the T category is subdivided from T4a to T1 (1.1%), T2 (5.7%), and T3a (15.5%), and from T4b to T3b (57.9%) and T4 (19.9%). Appendix A provides the survival curves stratified by T category with the 8th edition. The median OS values of the present patients in the T categories based on the 8th edition were as follows: T1, 12.5 months; T2, 10.9 months; T3a, 5.7 months; T3b, 3.9 months, and T4, 5.0 months. There was a significant difference in median OS between the patients with T2 and T3a tumors as well as between those with T3a and T3b tumors (Table 2). The 3-, 6-, and 12-month OS rates are given in Table 3. There were overlaps in survival rate between the patients with T1 and T2 tumors and between those with T3b and T4 tumors. No significant difference in the prognosis of the patients having T1 or T2 tumors, previously categorized together within T4a. The same was found between the patients with T3b and T4 tumors, previously categorized together in T4b.

Appendix A illustrates the OS values stratified by the patients’ N categories. The median OS of the N0 patients (5.9 months) was significantly better than that of the N1 patients (4.3 months, *p* = 0.00). The 3-, 6-, and 12-month survival rates of the N0 patients were 78.6% ± 2.8%, 49.0% ± 3.5%, and 27.9% ± 3.2%, and those of the N1 patients were 60.5% ± 2.2%, 38.2% ± 2.2%, and 18.1% ± 1.8%, respectively. Nodal involvement was commonly found (70%) in the ATC patients and was significantly more frequently (72.8%) identified among the patients with T3b or T4 tumors compared to those (61.9%) with T1, T2, or T3a tumors (*p* = 0.01)

The OS curves, according to the disease stage stratified by the 7th edition and the 8th edition, are provided in Figure 2. The median OS for the stage IV A patients improved from 10.6 (95% CI: 7.7–13.3) to 15.8 (8.5–27.6) months by applying the 8th edition, but the improvement was not significant (*p* = 0.17, log-rank). The median OS of the stage IV B patients did not change significantly, going from 6.0 (95%CI: 5.2–6.6) to 6.1 (5.7–6.8) months in the 8th edition (*p* = 0.48). No revision was made in the stage IV C category between the 7th and 8th editions, and the median OS was 2.8 (95% CI: 2.3–3.3) months. The 3-, 6-, and 12-month survivals of the patients in each stage are summarized in Table 4.

Sixty-three of 108 patients (58.3%) migrated from stage IV A of the 7th edition to the revised stage IV B of the 8th edition based on the existence of nodal involvement in their cases. These migrated patients showed significantly worse OS (8.7 months) compared to the revised stage IV A patients (*p* = 0.02). At the same time, the OS of these migrated patients was better than that of the former stage IV B patients (*p* = 0.03) (Figure 3).

A Cox proportional hazard model demonstrated that age ≥70 years old (vs. <70 yrs) and stage IV B or stage IV C (vs. stage IV A) were significant independent indicators of poor OS. The T4b category in the 7th edition also was revealed as one of the significant indicators of poor OS. The hazard ratios (HRs) for stage IV B and IV C against stage IV A were markedly increased (from 1.1 to 2.1 and from 2.5 to 4.5, respectively) in the 8th edition (Table 5 and Table 6).

## 3. Discussion

In the revised TNM classification, i.e., the 8th edition, ATC tumors are stratified into T categories as is done for DTC. Although a significant number of patients with DTC are re-classified into T1 or T2 and down-staged in the 8th edition [8], we observed no remarkable change in the classification of ATCs by applying the revised/8th edition TNM classification, and the T category was simply subdivided. There was no significant difference in prognosis between the patients with T1 and T2 tumors; they had all been stratified as T4a in the 7th edition. There was also no significant difference in prognosis between the patients with T3b and T4 tumors; those patients had been categorized as T4b in the 7th edition.

Our analyses revealed that the ATC patients with T3a tumors (i.e., tumors > 4 cm and limited to the thyroid) showed prognoses that were between those of the patients with T2 or T3b tumors, with significant between-group differences. These observations demonstrated that both large tumor size and extrathyroidal extension influenced the prognosis of the ATC patients, as has been reported [9,10,11,12,13]. Patients with T3a tumors should be treated carefully, even when they are classified as having stage IV A disease.

More than half (58%) of the patients who were classified as stage IV A in the 7th edition migrated to stage IV B in the 8th edition’s system due to the existence of nodal involvement. The influence of nodal involvement on the prognosis of ATC patients has not been evaluated in detail, due in part to the common nodal involvement in ATC cases and to the lack of the need for this information when the disease is classified by the 7th-edition TNM system. Only two studies were able to determine aspects of the importance of nodal involvement in the prognosis of ATC [1,13]. In the present study, we observed that nodal involvement was a significant indicator of poor prognosis, although it was not an independent factor. Our analyses revealed a significant correlation between the T category and the N category; nodal involvement was significantly frequently identified in patients with a T3b or T4 tumor. The patients who were classified as stage IV A by the 8th edition thus seem to have been selected patients who had clearly confined disease.

The revision of the TNM classification by the 8th edition reduced the number of present patients classified as stage IV A, and the proportion of these patients fell to only 6%. Instead, more than half of the patients were classified as stage IV B. Thus, the patients in the revised stage IV B could have a more heterogeneous disease compared to those in the former stage IV B. Additional information to determine therapeutic strategies should be collected for revised stage IV B patients, i.e., patients with locally advanced disease.

Surgery is the most effective method to manage ATC when a complete resection is possible. However, surgery is indicated in <20% of ATC patients in practice [1,2,14]. Extended surgery for patients with locally advanced disease has sometimes been effective to prolong survival, but the indications for this surgery require careful consideration due to the high rate of surgical complications that may ruin the quality of limited survival [15]. The Japanese Clinical Practice Guidelines [2] suggest the utility of the Prognostic Index (PI) for identifying patients who are appropriate candidates for aggressive treatment [9]. The PI is a simple system to stratify the prognoses of ATC patients at the time of the initial presentation. A good prognosis can be expected in ATC patients with no more than one positive factor of the following four poor prognostic indicators: (1) acute symptom within 1 month prior to presentation, (2) leukocytosis at ≥10,000 /mm^3^, (3) tumor > 5 cm, and (4) the existence of distant metastasis [9,16]. Figure 4. demonstrates the usefulness of the PI to determine the life expectancy of stage IV B patients. Significantly better survival was found in 237 patients with low PI (1 or less) compared with 185 patients with high PI (2 and more) (8.1 vs. 4.4 months, *p* = 0.00). The PI could be one of the useful indicators to plan an optimal therapeutic strategy, in addition to the TNM classification [1].

Many ATC patients develop recurrent disease within several months after even a curative operation at the local site. As shown by our present findings, the revision of the TNM system revealed that conservative treatment with surgery and radiation therapy [17] was not enough to cure ATC, especially in node-positive patients. Although multimodal treatment with surgery, radiation, and chemotherapy is the standard therapy of choice for ATC [2,3,17,18], no effective method has been identified to prevent recurrence. Novel therapeutic agents have been developed to control the systemic or inoperable disease of ATC, including paclitaxel [19], fosbretabulin [20], vascular endothelial growth factor receptor inhibitors: sorafenib [21] and lenvatinib (approved for ATC in Japan only) [22], selective tyrosine kinase inhibitors of V600E mutated BRAF gene: dabrafenib in combination with selective MEK (Mitogen-activated protein kinase/ Extracellular signal related kinase Kinase) inhibitor: trametinib (approved for ATC by the US Food and Drug Administration) [23,24], and immune checkpoint inhibitors [25,26]. In addition, successful neoadjuvant treatment with those agents followed by complete surgical resection as a novel multimodal therapeutic strategy was reported [24]. The prognoses of ATC patients will depend to some degree on the efficacies of these anticancer agents; in other words, the genetic alteration status of the tumor and/or the immune status of the patient may largely influence the patient’s prognosis. The predictors of therapeutic effects that were described herein should, therefore, be incorporated with the TNM system and the PI when considering individual therapeutic strategies for patients with ATC.

This study has several limitations. First, each patient was diagnosed individually in the participating institute, without central pathological confirmation. Second, patients with incidentally identified ATC were excluded due to their possible independent features differing from ordinally ATC [7]; these patients could additionally be included in stage IV A in the present series. Third, we could not distinguish T4a tumors from T4b tumors due to a lack of information in the accumulated patient data.

## 4. Materials and Methods

### 4.1. Patients

The case records of 1265 patients with ATC were accumulated in the ATCCJ database during the period from 2009 to 2019 from 57 institutions across Japan. We obtained the clinical factors, treatment information, and prognosis of each patient diagnosed as having ATC in a participating hospital by using REDCap (Research Electronic Data Capture) electronic data capture tools hosted at Osaka City University [27]. The dataset used for analysis was downloaded from the REDCap system on October 24, 2019. The ethics committee of Osaka City University approved this study (#3854).

### 4.2. Statistical Analyses

The statistical analyses were performed using EZR [28]. The differences in variables were examined using Fisher’s exact test. Kaplan–Meier curves of OS were created, and the differences in the OS were examined by the log-rank test. The impacts of the clinical factors, T classification, N classification, and TNM stage on the patients’ OS were evaluated by Cox proportional hazard models using a backward stepwise method. The relative risk for survival is presented as the hazard ratio (HR), 95% confidence interval (CI), and *p*-value < 0.05 were considered significant.

## 5. Conclusions

The revised AJCC/ UICC TNM classification system (8th edition) clearly indicates the prognoses of ATC patients. Patients with a subdivided T3a tumor showed prognoses that were between those of the patients with either a T2 or T3b tumor. More than half of the patients classified as stage IV A by the 7th edition migrated to stage IV B in the 8th edition, leaving a small proportion of patients with fair prognoses as having revised stage IV A disease and the majority of the ATC patients were categorized as stage IV B by revision, showing heterogeneous diseases.

## Figures and Tables

**Figure 1 cancers-12-00552-f001:**
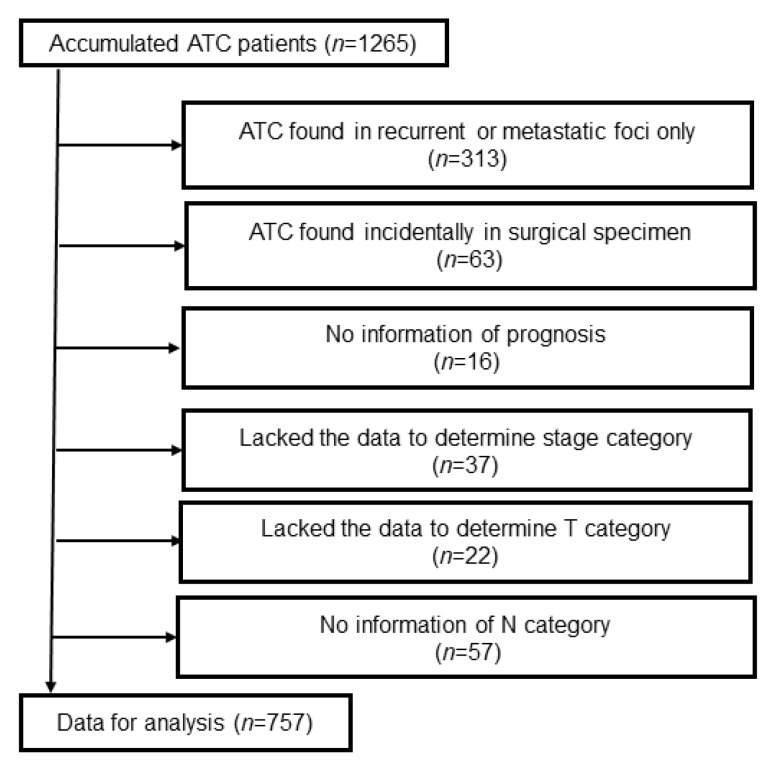
Cohort diagram.

**Figure 2 cancers-12-00552-f002:**
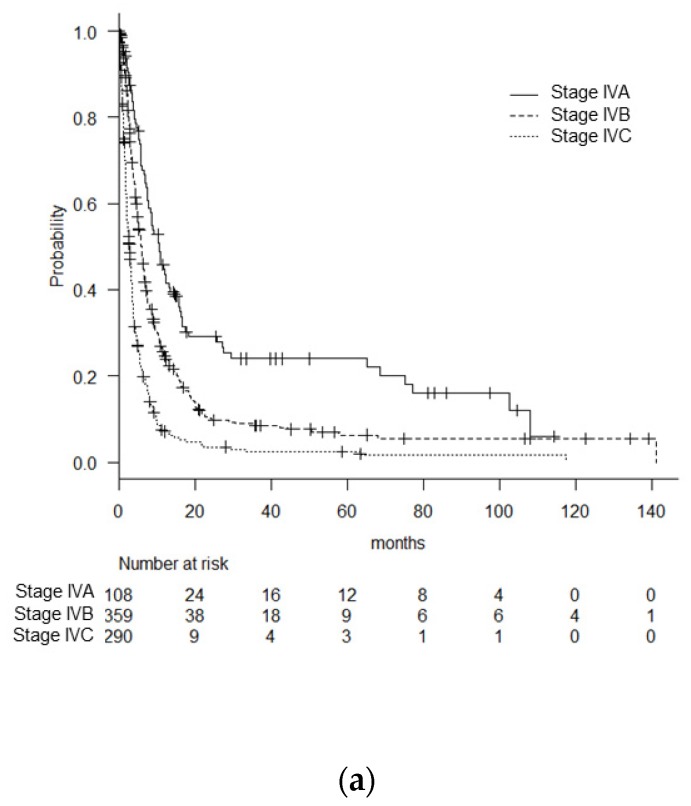
The OS of the patients according to the disease stage stratified by the American Joint Committee on Cancer/Union for International Cancer Control tumor–node–metastasis (AJCC/UICC TNM) classification, 7th edition (**a**) and 8th (**b**) edition.

**Figure 3 cancers-12-00552-f003:**
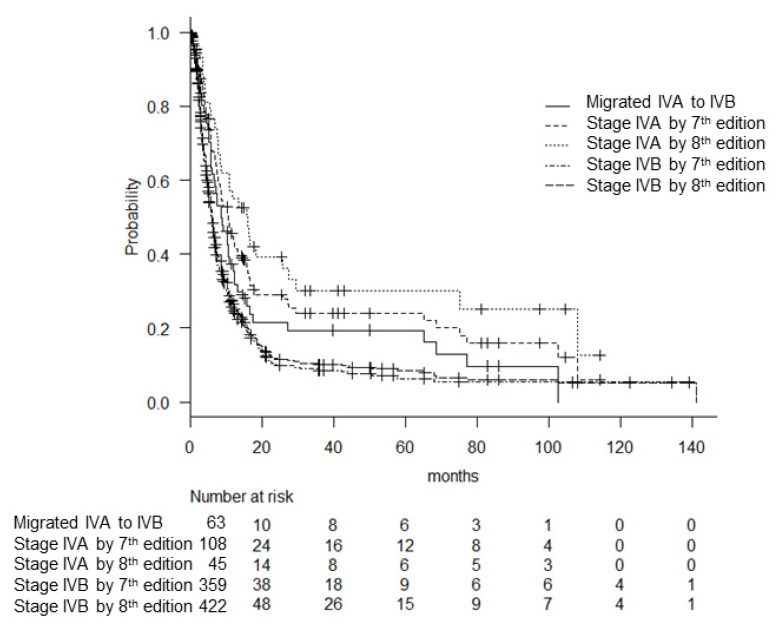
The patients who migrated from stage IV A of the 7th edition to stage IV B of the 8th edition showed significantly worse survival compared to the revised stage IV A patients (*p* = 0.02), as well as better survival than that of the former stage IV B patients (*p* = 0.03).

**Figure 4 cancers-12-00552-f004:**
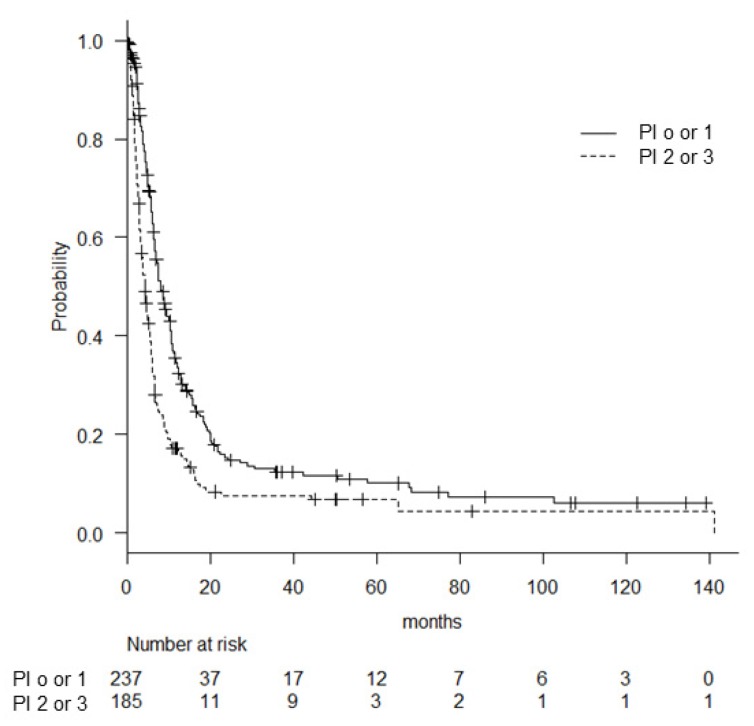
The overall survival (OS) of the patients with stage IV B disease stratified by the Prognostic Index.

**Table 1 cancers-12-00552-t001:** Revisions made by the 8th edition of the American Joint Committee on Cancer/Union for International Cancer Control tumor–node–metastasis (AJCC/UICC TNM) classification system of anaplastic thyroid cancer (ATC).

Stage	7th Edition	8th Edition
IV A	T4a, any N and M0 T4a: The cancer is any size but confined to the thyroid	T1–3a, N0 and M0 T1: Tumor ≤ 2 cm in greatest dimension limited to the thyroid T2: Tumor > 2 cm but ≤ 4 cm in greatest dimension limited to the thyroid T3a: Tumor > 4 cm limited to the thyroid
IV B	T4b, any N and M0 T4b: The cancer is extended outside of the thyroid gland to any extent	T1–3a, N1 and M0 Or T3b–T4b, any N and M0 T3b: Gross extrathyroidal extension invading only strap muscles (sternohyoid, sternothyroid, thyrohyoid, or omohyoid muscles) from a tumor of any size T4a: Gross extrathyroidal extension invading subcutaneous soft tissues, larynx, trachea, esophagus, or recurrent laryngeal nerve from a tumor of any size T4b: Gross extrathyroidal extension invading prevertebral fascia or encasing a carotid artery or mediastinal vessels from a tumor of any size
IV C	Any T, any N and M1	Any T, any N and M1

**Table 2 cancers-12-00552-t002:** Distribution and overall survival of the ATC patients according to the clinical factors and AJCC/UICC TMN classification.

Factor		No.	%	Median Overall Survival Months 95%CI (Range)	Log-Rank Test *p*-Value
Age:		Median 71.0 (28–96) yrs			
	<70 yrs	349	46.1%	5.0 (4.3–6.0)	0.01
	≥70 yrs	408	53.9%	4.6 (3.8–5.3)	
Sex:					
	Male	305	40.3%	4.9 (3.8–5.8)	0.74
	Female	452	59.7%	4.6 (4.1–5.4)	
T (7th edn.):					
	4a	168	22.2%	7.5 (5.7–8.9)	< 0.01
	4b	589	77.8%	4.3 (3.7–4.7)	
T (8th edn.):					
	1	8	1.1%	12.5 (0.9–16.8)	n.s. ^1^ vs. others
	2	43	5.7%	10.9 (8.2–17.7)	0.03 vs. T3a <0.01 vs. T3b and T4
	3a	117	15.5%	5.7 (4.3–7.3)	<0.01 vs. T3b 0.03 vs. T4
	3b	438	57.9%	3.9 (3.4–4.5)	0.66 vs. T4
	4a and 4b	151	19.9%	5.0(3.8–6.4)	
N:					
	0	224	29.6%	5.9 (4.8–7.0)	<0.01
	1	533	70.4%	4.3 (3.7–4.9)	
M:					
	0	467	61.7%	6.6 (5.9–7.3)	<0.01
	1	290	38.3%	2.8 (2.3–3.3)	
Stage (7th edn.):					
	IV A	108	14.3%	10.6 (7.7–13.3)	<0.01 between each
	IV B	359	47.4%	6.0 (5.2–6.6)	
	IV C	290	38.3%	2.8 (2.3–3.3)	
Stage (8th edn.):					
	IV A	45	5.9%	15.8 (8.5–27.6)	<0.01 between each
	IV B	422	55.7%	6.1 (5.7–6.8)	
	IV C	290	38.3%	2.8 (2.3–3.3)	

^1^ n.s.: not significant.

**Table 3 cancers-12-00552-t003:** The 3-, 6-, and 12-month overall survival (OS) rates of the ATC patients according to T category stratified by the AJCC/UICC TNM classification 8th edition.

T	3 Mos.	6 Mos.	12 Mos.
1	71.4% ± 17.1%	71.4% ± 17.1%	42.9% ± 18.7%
2	80.2 % ± 6.3%	72.7% ± 7.0%	47.1% ± 8.0%
3a	74.7% ± 4.1%	45.7% ± 4.7%	25.9% ± 4.2%
3b	61.4% ± 2.4%	35.7% ± 2.4%	17.1% ± 1.9%
4	65.1% ± 4.0%	41.6% ± 4.3%	16.9% ± 3.3%

The estimated OS rates ± standard error are shown.

**Table 4 cancers-12-00552-t004:** The 3-, 6-, and 12-month OS rates of the ATC patients according to disease stage stratified by the 7th or 8th edition AJCC/ UICC TNM classification.

Stage	3 Mos.	6 Mos.	12 Mos.
IV A, 7th edn.	86.5% ± 3.3%	67.7% ± 4.6%	43.6% ± 5.0%
IV B, 7th edn.	74.5% ± 2.4%	48.8% ± 2.7%	24.6% ± 2.4%
IV A, 8th edn.	90.7% ± 4.4%	74.0% ± 6.8%	52.5% ± 7.7%
IV B, 8th edn.	75.9% ± 2.1%	50.8% ± 2.5%	26.6% ± 2.3%
IV C, 7th and 8th edn.	46.0% ± 3.0%	21.1% ± 2.5%	6.6% ± 1.6%

Estimated OS rates ± standard error are shown.

**Table 5 cancers-12-00552-t005:** Multivariate analysis of the factors influencing overall survival of the ATC patients according to the AJCC/ UICC TNM classification, 7th edition.

Factors	Univariate			Multivariate		
	HR	95% CI	*p*	HR	95% CI	*p*
Age < 70 vs. ≥ 70 yrs.	1.3	1.1–1.5	0.002	1.3	1.1–1.5	0.002
Male vs. female	1.1	0.9–1.3	0.323			
T4b vs. T4a	1.5	1.1–2.1	0.008	1.6	1.3–2.1	0.004
N1 vs. N0	1.2	1.0–1.5	0.016			
M1 vs. M0	2.5	1.7–3.5	0.000			
Stage IV B vs. IV A	1.1	0.7–1.6	0.688	1.1	0.73–1.6	0.744
Stage IV C vs. IV A	NA ^1^	NA–NA	NA	2.5	1.73–3.5	< 0.001

^1^ NA: not available. HR: hazard ratio.

**Table 6 cancers-12-00552-t006:** Multivariate analysis of the factors influencing overall survival of the ATC patients according to the AJCC/ UICC TNM classification, 8th edition.

Factors	Univariate			Multivariate		
	HR	95% CI	*p*	HR	95% CI	*p*
Age <70 vs. ≥ 70 yrs.	1.3	1.1–1.6	0.001	1.3	1.1–1.5	0.002
Male vs. female	1.1	0.9–1.3	0.306			
T2 vs. T1	0.6	0.3–1.4	0.273			
T3a vs. T1	1.0	0.5–2.2	0.934			
T3b vs. T1	1.4	0.7–3.0	0.369			
T4b vs. T1	1.2	0.5–2.5	0.703			
N1 vs. N0	1.2	1.0–1.5	0.030			
M1 vs. M0	3.0	1. 9–4.6	0.000			
Stage IV B vs. IV A	1.3	0.8–2.0	0.279	2.1	1.4–3.0	< 0.001
Stage IV C vs. IV A	NA ^1^	NA–NA	NA	4.5	3.0–6.6	< 0.001

^1^ NA: not available. HR: hazard ratio.

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
