# Peer review of "Evaluation of the 8th Edition TNM Classification for Anaplastic Thyroid Carcinoma"

_cancers, 2020, doi:10.3390/cancers12030552_

Round 1
Reviewer 1 Report
The authors present an interesting study of the current UICC staging system in comparison to the previous 7th edition for anaplastic thyroid carcinoma (ATC).
The paper is overall well written and the results are presented in a clear manner.
The major finding is that there is a shift in stage from IVA to IV for patients with lymph node involvement in a substantial proportion of patients previously assigned to stage IVA.
I have only minor comments:
table 1: copy/paste error in stage IVC: IV C Ant T, any N and M1 Ant T, any N and M1 instead of IV C Any T, any N and M1 Any T, any N and M1 line 90: 63 cases had an ATC lesion that was91 incidentally identified after the pathological investigation of the surgical specimen: I do not see why these were not included into the analysis. The authors should compare the prognosis incidentally found ATC to those with corresponding stage not found incidentally all tables: p=0.00 from my view should be <0.001 line 108: There were overlaps in survival rate between the patients with T1 and 109 T2 tumors and between those with T3b and T4 tumors
Please elaborate on this line 200 and following: the section on therapeutic approaches should be expanded. For instance it should be mentioned that lenvatinib is approved for ATC but only in Japan, and of trametinib and dabrafenib in the US. The authors also should comment on the value of multimodal therapy in stage IVC (e.g. Wendler et al., Eur J Endocrinol) and the only fosbretabulin phase III study (Sosa et al Thyroid 2014) to allow the reader to better appreciate the therapeutic options. NTREK should be NTRK and the authors should verify whether this has been indeed applied in ATC (in the Drilon paper I don't find information about the subtype) Line 230: the authors should include the type of model (forward/backward inclusion etc)
Author Response
Thank you so much for your detailed review and genorous comment. We tryed to revise our manuscript according to your kind suggestions as below.
- table 1: copy/paste error in stage IVC: IV C Ant T, any N and M1 Ant T, any N and M1 instead of IV C Any T, any N and M1 Any T, any N and M1.
>We are so sorry for the error. We correct them.
- line 90: 63 cases had an ATC lesion that was incidentally identified after the pathological investigation of the surgical specimen: I do not see why these were not included into the analysis. The authors should compare the prognosis incidentally found ATC to those with corresponding stage not found incidentally.
>Thank you for your precious comment. We have previously reported the characteristics and prognosis of the incidentally found ATC in a paper by Yoshida A. et al. (ref 7) independently from ordinally ATC. As described in the paper, incidentally identified ATC had particular features. They often associated with concomitant papillary carcinoma and showed more favorable prognosis compared with ordinally ATC. Moreover, in those case, 1) accurate diameter of ATC might be difficult to measure, 2) nodal involvement and/ or distant metastasis could often be from differentiated carcinoma, 3) preoperative or operative findings might be due to coexisting DTC and be inappropriate to describe the characteristics of ATC. Furthermore, some of these were found only within the metastatic lesion. These characteristics made us to exclude incidentally identified ATC in this study: focusing on the evaluation of the revision of TNM classification. We added some comment as in line 93-99 and 226-228 to describe these reasons.
- all tables: p=0.00 from my view should be <0.001.
>Thank you for the comment. We corrected them.
- line 108: There were overlaps in survival rate between the patients with T1 and 109 T2 tumors and between those with T3b and T4 tumors Please elaborate on this.
>Thank you for the detailed review. We added some comment to clarify the meaning in line 115 to 117.
- line 200 and following: the section on therapeutic approaches should be expanded. For instance it should be mentioned that lenvatinib is approved for ATC but only in Japan, and of trametinib and dabrafenib in the US. The authors also should comment on the value of multimodal therapy in stage IVC (e.g. Wendler et al., Eur J Endocrinol) and the only fosbretabulin phase III study (Sosa et al Thyroid 2014) to allow the reader to better appreciate the therapeutic options. NTREK should be NTRK and the authors should verify whether this has been indeed applied in ATC (in the Drilon paper I don't find information about the subtype).
> We appreciate your precious comment on the future directions of the treatment of ATC. We tried to expand our description adding some references according to your recommendation as in line 209-220. We also revised our description and deleted NTRK according to your generous comments.
- Line 230: the authors should include the type of model (forward/backward inclusion etc).
>We use backward stepwise method. We added comments.
Reviewer 2 Report
The authors evaluated the new TNM classification system in a large patient database (n=757). They found the revised TNM system useful in patient selection and determining patients´prognosis e.g. 58% of all patients classified as stage IV A by the 7th edition migrated to stage IV B in the 8th edition. As a result, stage IV A according to the revised TNM system represents a patient cohort with a fair prognoses.
The manuscript is well written and highlights the importance of the revised TNM classification system into clinical practise. The manuscript is timely, and of interest to the readership of Cancers. However, the work suffers from some limitations: absence of central pathological confirmation, differentation of T4a tumors from T4b tumors due to lack of information
The authors suggest that incidentally diagnosed ATC could be included in stage IV A in the present series. Please comment and describe in more detail.
In summary, short but nicely written manuscript of an important aspects for all oncologists, including of its significantly appropriate timing.
Author Response
Thank you so much for your detailed review and genorous comment. We will go forward for novel projects with our nationwide consortium.
The authors suggest that incidentally diagnosed ATC could be included in stage IV A in the present series. Please comment and describe in more detail.
>Thank you for your precious comment. We have previously reported the characteristics and prognosis of the incidentally found ATC in a paper by Yoshida A. et al. (ref 7) independently from ordinally ATC. As described in the paper, incidentally identified ATC had particular features. They often associated with concomitant papillary carcinoma and showed more favorable prognosis compared with ordinally ATC. Moreover, in those case, 1) accurate diameter of ATC might be difficult to measure, 2) nodal involvement and/ or distant metastasis could often be from differentiated carcinoma, 3) preoperative or operative findings might be due to coexisting DTC and be inappropriate to describe the characteristics of ATC. Furthermore, some of these were found only within the metastatic lesion. These characteristics made us to exclude incidentally identified ATC in this study: focusing on the evaluation of the revision of TNM classification. We added some comment as in line 93-99 and 226-228 to describe these reasons.
Reviewer 3 Report
This is an interesting paper showing the experience concerning anaplastic thyroid carcinoma (ATC) from 57 institutions across Japan. Their findings support that the new (8th) edition TNM classification better stratifies the prognosis of patients with ATC. They also illustrate the usefulness of the Prognostic Index based on the Japanese Clinical Guidelines to identify patients who are appropriate candidates for aggressive treatment.
Author Response
Thank you so much for your detailed review and genorous comment. We will go forward for novel projects with our nationwide consortium.